# Something Happened with the Way We Work: Evaluating the Implementation of the Reducing Coercion in Norway (ReCoN) Intervention in Primary Mental Health Care

**DOI:** 10.3390/healthcare12070786

**Published:** 2024-04-04

**Authors:** Tonje Lossius Husum, Irene Wormdahl, Solveig H. H. Kjus, Trond Hatling, Jorun Rugkåsa

**Affiliations:** 1Faculty of Health Sciences, Oslo Metropolitan University, 0166 Oslo, Norway; jorunrug@oslomet.no; 2Department of Mental Health Work, NTNU Social Research, 7491 Trondheim, Norway; irene.wormdahl@samforsk.no; 3Norwegian Resource Centre for Community Mental Health, NTNU Social Research, 7491 Trondheim, Norway; solveig.kjus@samforsk.no (S.H.H.K.); trond.hatling@samforsk.no (T.H.); 4Health Services Research Unit, Akershus University Hospital, 1478 Lørenskog, Norway; 5Centre for Care Research, University of South-Eastern Norway, 3918 Porsgrunn, Norway

**Keywords:** reducing coercion, involuntary admission, process evaluation, implementation research, co-creation, complex intervention, mental health services, participatory research, primary mental health care

## Abstract

Background: Current policies to reduce the use of involuntary admissions are largely oriented towards specialist mental health care and have had limited success. We co-created, with stakeholders in five Norwegian municipalities, the ‘Reducing Coercion in Norway’ (ReCoN) intervention that aims to reduce involuntary admissions by improving the way in which primary mental health services work and collaborate. The intervention was implemented in five municipalities and is being tested in a cluster randomized control trial, which is yet to be published. The present study evaluates the implementation process in the five intervention municipalities. To assess how the intervention was executed, we report on how its different elements were implemented, and what helped or hindered implementation. Methods: We assessed the process using qualitative methods. Data included detailed notes from quarterly progress interviews with (i) intervention coordinators and representatives from (ii) user organisations and (iii) carer organisations. Finally, an end-of-intervention evaluation seminar included participants from across the sites. Results: The majority of intervention actions were implemented. We believe this was enabled by the co-creating process, which ensured ownership and a good fit for the local setting. The analysis of facilitators and barriers showed a high degree of interconnectedness between different parts of the intervention so that success (or lack thereof) in one area affected the success in others. Future implementation should pay attention to enhanced planning and training, clarify the role and contribution of service user and carer involvement, and pay close attention to the need for implementation support and whether this should be external or internal to services. Conclusions: It is feasible to implement a complex intervention designed to reduce the use of involuntary admissions in general support services, such as the Norwegian primary mental health services. This could have implications for national and international policy aimed at reducing the use of involuntary care.

## 1. Introduction

Public policy in a number of countries seeks to reduce the use of involuntary psychiatric admissions [1,2]. Calls for such reduction come from international bodies, including the UN, WHO, and Council of Europe [2,3,4], and service user organisations [5]. Nonetheless, levels of involuntary admissions seem to remain stable or increase across countries [6], including countries with explicit ambitions for reduction. This is of concern because involuntary psychiatric admissions restrict personal autonomy and are, by some, considered to conflict with human rights [7].

Involuntary admissions are sanctioned by mental health legislation. Authority to impose them is usually vested in psychiatrists (and in some places, approved psychologists) working in secondary or tertiary care, with their hospital as the legal entity. This is probably why policies to reduce involuntary care, in general, are directed towards the specialist care level. In Norway, for instance, it is the regional health authorities responsible for specialist care that are instructed by the Government to ensure reduced and appropriate use of involuntary care in their area [8]. At the same time, an emphasis on community-based care and the scaling down of hospital beds [9] means most people who are admitted involuntarily live their lives and receive their services in a community setting. It is only when their condition becomes acute, usually involving a risk to their life or health, that they are referred to involuntary admissions. Policies to reduce involuntary admissions are therefore premised on the ability of specialist services to find ways to avoid the compulsion of people in an acute psychiatric crisis. To address this apparent mismatch between policy and service configuration, we report on a study that takes a different approach, in that we examined whether the support, care, and services people receive at the lowest level of service delivery might help them remain well and thus prevent involuntary admissions. We direct our focus away from ‘crisis care’ towards what Gooding and colleagues have referred to as ‘general support services’ [10]. Crisis-oriented out-patient interventions to reduce coercive care (e.g., crisis planning, risk assessment, and crisis residential care) show promising results [11], as do ACT and FACT models that avoid involuntary care through ongoing support [12]. These services are, however, usually not available to everyone, and where they exist, they often depend on referrals for specialist care.

We have not found any published interventions to reduce involuntary admission aimed at the lowest level of care and accessible to all. We, therefore, together with stakeholders in five municipalities, created the Reducing Coercion in Norway (ReCoN) intervention [13]. In Norway, it is municipalities (local authorities) who are responsible for providing primary health and social care to their residents. The aim was, within existing resources, to change the way primary mental health services work and collaborate with local partners, so as to reduce the need for involuntary admissions. The intervention is being tested in a cluster-randomised controlled trial, the results of which are forthcoming. Five municipalities were randomised to develop, and subsequently implement the invention over 18 months, and five were randomised to control (ClinicalTrials.gov ID: NCT03989765). By design, both arms had rates of involuntary admissions above the national average.

This article presents an evaluation of the implementation of the ReCoN intervention. The purpose is to assess its feasibility and success in its service context and to inform the interpretation of trial results. Such a contextual approach is recommended when evaluating complex interventions (i.e., more than one ‘active ingredient’ working simultaneously). This is particularly important in studies of initiatives to reduce coercion, as it might help explain why existing policy initiatives often fail [14]. Specifically, our objectives were to investigate (i) to what extent the different elements of the intervention were implemented, and (ii) what helped or hindered implementation. Based on these results and the wider literature, we then consider, in the Discussion Section how the intervention should be modified or adapted for future implementation.

### 1.1. The Service and Legal Context

Norway is a high-income country with extensive public welfare services. Primary care, including mental health care, is the responsibility of the 356 municipalities. Multi-disciplinary primary mental health teams, usually organised together with addiction services, provide long-term follow-up (often for many years) of people with severe mental health problems. This can include supported housing (with or without resident staff), daycare facilities, domestic support, supported leisure activities, transport to service appointments, administration of medication, and therapeutic conversations. Municipalities must ensure that inhabitants have access to a General Practitioner (GP/family doctor) and accident and emergency services (A&E). The municipalities also administer social care, social security benefits, (un)employment services, and housing services.

Specialist mental health care is the responsibility of four Regional Health Authorities that, through 20 Hospital Trusts, provide inpatient treatment (acute wards, high-security wards) and Community Mental Health Centres (CMHC) with specialist community-based inpatient and outpatient services. Municipalities are nested within catchment areas of a total of 65 CMHCs, and the two levels are expected to collaborate closely. In practice, this is not always easy to achieve [15].

Involuntary psychiatric admissions are regulated by the Norwegian Mental Health Act [16]. The legal criteria include the presence of a severe mental disorder (or, for involuntary observation, suspected disorder), and either that the person’s prospect for improvement will be considerably reduced if they do not receive treatment (the treatment criterion), or that the person poses an immediate and serious risk to their own life or the life or health of others (the dangerousness criterion). Compulsory care is restricted to patients assessed to lack the capacity to consent to treatment (the capacity criterion) unless the dangerousness criterion is met. All options for voluntary engagement must have been exhausted, and the patient has the right to be heard. A first assessment for involuntary admission is made by a doctor independent of the admitting hospital, usually a GP or A&E doctor. If the individual refuses assessment, the Chief Municipal Medical Officer (CMMO) can authorise that they are assessed involuntarily. When a referral for involuntary admission is made, the person is transported, sometimes by the police, to an acute inpatient hospital ward. Here, a separate medical assessment must confirm the decision within 24 h.

### 1.2. The ReCoN Intervention and Implementation Plan

A full outline of the co-creation process is available elsewhere [17]. In brief, we first mapped current practices and common features of pathways ending in involuntary admissions. This was done through qualitative interviews across the five intervention sites, with a sample of 103 stakeholders that included people with lived experience of severe mental illness, some of whom had experienced involuntary admissions; family carers; municipal mental health professionals; managers of municipal services; GPs and A&E doctors; specialist CMHT workers, and police officers. Results from this mapping exercise [18,19] were presented at dialogue conferences [20,21] in each site, in which representatives from the stakeholder groups took part. The purpose was to identify achievable measures by which the primary mental health service, in collaboration with partners, could impact the course of pathways that often end in involuntary admissions. We also presented, to inform and inspire, the Six Core Strategies [22,23] as an example of a successful, comprehensive intervention shown to reduce the use of restraints and seclusion of inpatients. Several recent reviews have concluded that, given the complexities surrounding involuntary care, interventions that include multiple components are likely to have the greatest impact [11,24]. Consequently, a wide range of measures were assessed for their potential to prevent involuntary admission while also being considered implementable. Each conference produced a prioritised list of action points. These were taken forward in a set of iterative discussions between local stakeholders and the research team, through which the intervention was finalised [17]. As shown in Table 1, the intervention consists of six strategy areas, each of which contains two to four areas of action with a number of specific action points (hereafter ‘actions’), 53 in total. The complexity and ambition of the intervention resulted from the motivation of participants. Given the breadth of the actions that were considered suitable, some of them were already fully or partially in place at some sites. Collaboration with two national organisations for service users (Mental Health Norway) and carers (Mental Health Carers Norway (LPP)), which have local branches in many municipalities nationally, formed the backbone of the strategy for involving users at the strategic level.

The original intervention period was 12 months from 1 October 2020. As a result of COVID-19 restrictions, some actions took longer than anticipated to implement, and on the initiative of municipal partners, the period was extended by six months. A structured plan for consolidating and supporting the implementation was drawn up jointly by the municipal services and the research team and contained the following items, the timeline for which can be seen in Figure 1:In each site, two or three staff from the municipal mental health service were allocated the role of ‘ReCoN coordinators’. An informal working group was established in each municipality. These consisted of coordinators plus the service managers considered appropriate, so varied between sites. In one site, representatives from the user and carer organisations were included.A detailed implementation manual was produced by the researchers. This included the rationale and evidence for each strategy area and details of actions with specified responsibilities and suggested timelines [17]. It also contained templates for post-incident reviews and joint crisis plans, to be used or adapted for strategy area 2.A kick-off seminar was organized by coordinators in each municipality, to which relevant stakeholders were invited. The aim was to mark the start of the intervention period and to create enthusiasm and ownership locally.Implementation meetings between coordinators across the five municipalities and researchers took place online every two months to discuss progress. The nine meetings focused on ‘problem-solving’ particular actions that someone struggled with through discussion and exchange of experiences across sites.Separate implementation meetings were also held between the research team and local users and organisations, respectively. A total of 13 meetings focused on users and carers involvement in implementation, and the experiences captured were fed into subsequent implementation meetings with coordinators.Training courses were arranged by the research team on recovery-oriented services and trauma-informed care to support professional development as part of Strategy Area 3. The latter included a module on implementation issues aimed at managers. Training in the assessment of decision-making capacity was available online.A ReCoN newsletter was produced by the researchers and circulated by email every three months, featuring information about implementation progress and national research on involuntary care. A Facebook group also kept those interested updated.At the 12-month point, seminars aimed to take stock of progress and boost motivation for the last six months were held in each site, to which all stakeholders were invited.

**Figure 1 healthcare-12-00786-f001:**
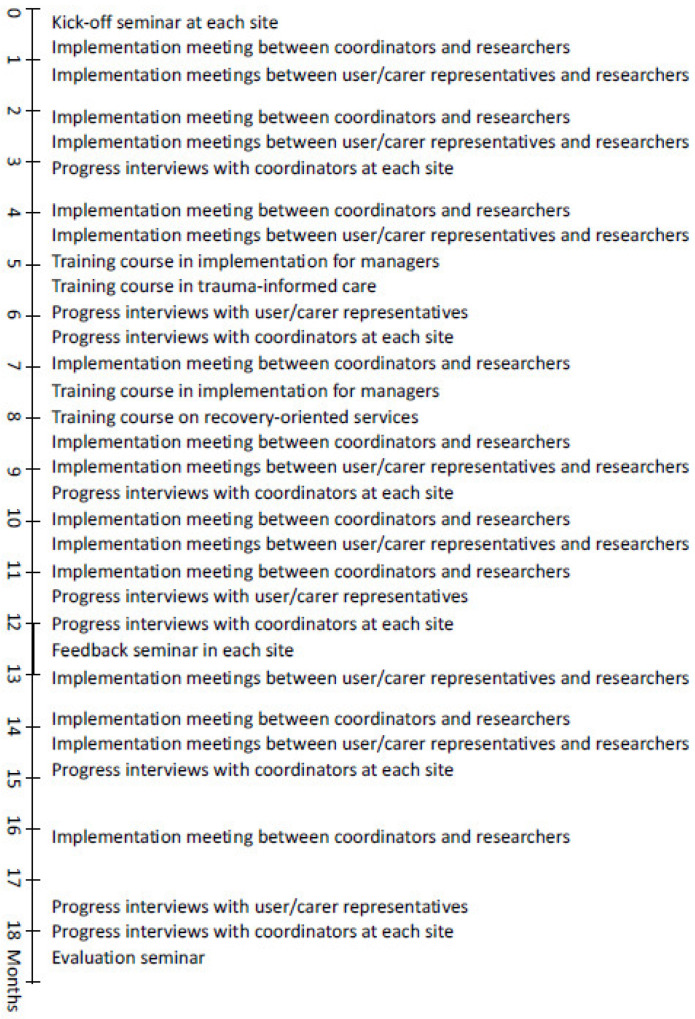
Timeline of the activities during the implementation period.

## 2. Materials and Methods

### 2.1. Design and Data Collection

The data on which the below analysis is based were extensive notes from 45 progress interviews and the evaluation seminar. As shown in Figure 1, quarterly progress interviews with coordinators in each site were conducted at 3, 6, 9, 12, 15, and 18 months. We explored the coordinators’ perspectives on each of the 53 actions and discussed solutions to potential difficulties. At 12 and 18 months we asked specifically about facilitators and barriers. IW and TLH divided the sites between them but conducted the first interview in each round jointly to ensure consistency. Each of the 30 interviews lasted around two hours. Once an action was considered fully implemented, it was not assessed subsequently at that site. Progress interviews with local user and carer organisations, along with peer workers where these were in place, were conducted in each site at 6, 12, and 18 months by SHK and TH. The 15 interviews followed the same format as those with the coordinators.

An ‘end of intervention’ evaluation seminar with coordinators and representatives from user and carer organisations from across the five sites, was held in May 2022, with approximately 25 participants. Here, experiences were shared about actions that worked or not, potential reasons why and suggested improvements to intervention design. While progress interviews were conducted digitally due to COVID-19 restrictions, the evaluation seminar had physical attendance.

To assess the degree of implementation, we categorised answers from the qualitative interviews to indicate whether actions (or combination of actions where they followed each other) were fully implemented (scored as 3), partly implemented (as 2), or not implemented (as 1). IW and THL assessed all data twice to check consistency in the scoring, making adjustments in the event of inconsistencies, and resolving differing interpretations through discussion. As we return to below, user representatives expressed not having been sufficiently involved to comment on a number of actions. This also meant that scores could not include their perspectives, except on actions surrounding user involvement.

### 2.2. Analysis

To answer our first research question, the categorisation of the degree of implementation was plotted in Excel. As these data were qualitative in nature, they provided an impression of trends in what was implemented (or not) and allowed for crude comparisons between sites through graphic depiction.

The second research question was addressed through a thematic analysis [26], in inductive-deductive cycles of the data. We first took a largely deductive approach to identify enablers and barriers to implementation as experienced by participants. Separate analyses of interviews with coordinators (IW and TLH) and user/carer representatives (SHK and TH) were conducted. All data was assessed numerous times in a series of meetings of the research team to identify overarching themes that applied across the intervention. We found that these themes overlapped with strategy areas so that the presence or absence of one theme (e.g., Leadership) impacted other areas. Second, we sorted data under each strategy area, and subsequently each action, before conducting a descriptive constant comparison analysis [27] to ensure both similar and discrepant experiences of what helped and hindered implementation across sites were accounted for.

Below, we use excerpts from the data to illustrate and validate interpretations [28]. Unless otherwise specified, these are selected as they represent shared views and are only identified by municipality number [1,2,3,4,5], to protect anonymity.

### 2.3. Ethics

The Regional Committees for Medical and Health Research Ethics deemed the study to fall outside the remit of the Norwegian Health Research Act (ref: 2018/2382 C). The protocol was approved by the Norwegian Centre for Research Data (ref: 743586) and the study was carried out in strict accordance with that approval. This included obtaining informed consent from informants who were interviewed, and that no identifiable personal information is published.

## 3. Results

### 3.1. To What Extent Was the Intervention Implemented?

The coordinators reported that overall, most of the actions were implemented fully or partly. Figure 2 shows the combined scores in each site (0–100% implementation) over time. Three sites reached an implementation degree of around 90%, while two ended between 70–75%. Most actions were achieved during the first 12 months, though there were developments across the entire period.

None of the five municipalities completed all actions, but all were implemented in at least one area. There were no clear patterns across sites of which actions were implemented, to what degree, and when. The strategy area with the biggest difference between municipalities was area 4 on collaboration between service levels. In one municipality, structured arrangements for this were already in place, while another reported struggling to make progress.

Participants expressed that implementation was slower than anticipated. This was often with reference to COVID-19 restrictions and the need to switch to digital modes of interaction. The impression was that the extra six months helped them implement the actions more fully.

### 3.2. What Helped or Hindered Implementation?

#### 3.2.1. Overarching Themes

Several prolonged periods of COVID-19 lockdowns formed the backdrop to the implementation. This affected all six strategy areas in that movement and social interactions were severely limited, and infection control put strain on all services. This particular affected the CMMOs’ and local GPs’ ability to engage in implementation. In addition, sick leave and the use of temporary or inexperienced staff increased across services:


*“The pandemic has had a huge negative effect because we could not meet in person… Collaborating with other services has been much harder. All the work with vaccination and contact tracing has taken a lot of the time and resources of everyone working in municipal health services.”*
M2

With most actions being achieved, the coordinators nonetheless expressed that implementation had largely been successful. Many emphasised that the implementation plan, and particularly the manual and the continuous contact with and support from the research team, had been key to sustaining momentum:


*“The regular and close follow-up from the research team throughout the implementation has been a success factor. It’s helped the project keep momentum.”*
M4

Some explained that the systematic approach of the intervention had been mainstreamed into organisational planning for future work:


*“The systematic way of working that ReCoN entails has been good, and this has spill-over effects into other parts of our work too. For example, our future plans narrow focus down to 2–3 areas that we will focus properly on”*
M5

Coordinators expressed that while collaboration both within and between service levels had improved, there were still some difficulties in that regard, with impact across strategy areas. They also emphasized, as did some user and carer representatives, that there had been a shift in attitudes during the intervention period so that they were now more concerned about, and better at helping, those at risk of involuntary admissions:


*“We are now cooperating more closely within the municipality and with specialist services, to stabilize the everyday lives of users when they live at home, in order to break patterns of frequent compulsory hospitalizations… This has a lot to do with attitudes to act in ways other than simply admitting.”*
M3

In hindsight, some said that they might have been overly ambitious when co-creating the intervention and that a stripped-down version might have been more manageable. ‘Real life’ issues other than the pandemic had affected implementation, such as simultaneously being involved in other projects, or ‘reorganisation fatigue’ resulting from consecutive changes to service organisations. Some said they had underestimated how much time was required to try to implement a complex intervention throughout the local services. Moreover, some expressed the ambition to reduce the use of compulsory care was not shared by all. Some members of staff were described as afraid of those with histories of violence or heavy drug use, and that their emphasis on controlling risk could be contrary to a focus on reducing coercion.

#### 3.2.2. Strategy Area 1: Management

A key aim of Strategy Area 1 was to anchor the intervention at the managerial level, to ensure strategic support for implementation. To achieve this, anchoring at a relatively high organisational level was considered advantageous. Having ReCoN as a regular topic at management meetings facilitated the information flow to those working in and across collaborating services. A downside to anchoring at a high level was the distance to the day-to-day work and the target group:


*“We should have anchored the project differently. We anchored it at a high managerial level, but they were too far removed from service delivery, staff, and users. We did not achieve stable anchorage at lower managerial level … which has meant that it has been difficult to create enthusiasm and commitment among staff, and to prioritize systematic work with intervention across departments.”*
M4

Appointing local coordinators was another means of anchoring the intervention. This action was completed early across sites and was seen as essential for the progress that followed. Again, different concerns needed to be weighed up. If the coordinators were too high up in the organisation, they might be “out of touch” with actual service delivery, yet if they were too low, they might lack sufficient overview or authority to stimulate change. The role was therefore usually given to someone in mid-management.

Among actions to support leadership were setting up structures for obtaining monitoring data and using that to assess progress. Data on how many, and who, in a case mix were subjected to involuntary care were not routinely held by the municipal services and needed to be obtained from the local hospital. Some sites obtained this easily, while in others, the hospital had limited resources to extract tailored data, or were concerned over the legality of data sharing. When data was obtained, it was described as a valuable tool to gain an overview, monitor the situation, and sometimes alert services to those at risk of involuntary admission and therefore needed targeted care (see Strategy area 6).


*“The monitoring figures and the “hands-on” from the top are crucial factors. What would have happened to some of these people if we hadn’t known about them? Now we have an overview that we didn’t have before, and we have been able to deal with more holistic follow-up, housing, and treatment for those where we identify need.”*
M3

As this quote suggests, the use of monitoring data was experienced as valuable for evaluating progress and also for the implementation of other strategy areas (see below). The actions of regularly evaluating and reflecting on situations that ended with a referral for involuntary admission were reported as useful. Sharing successful stories at regular management meetings of how these situations were addressed, was similarly described as useful. Such sharing was found to ensure progress through learning and by maintaining motivation:


*“Focusing on the project’s stories of success has in itself been a success story. We share success stories [from clinical practice with the target population] as a regular part of the agenda of management meetings. It provides motivation and stimulus… it promotes learning of what different services do and… gives recognition for the work you do.”*
M5

#### 3.2.3. Strategy Area 2: Involving Persons with Lived Experience and Family Carers

There were actions for user and carer involvement throughout the planning, implementation, and evaluation stages. Coordinators reported that the action of inviting these representatives to be involved, and to training events, had largely been accomplished. Coordinators at all sites described some involvement at the strategic level of implementation, such as regular meetings with the Municipal User Council or consultation with representatives from the local user- and/or carer advocacy organisations. Two sites reported difficulties in maintaining user involvement over time, due to the pandemic or changes of personnel in the identified local user/carer organisations. Some coordinators reported that the actions involving user representatives in assessing progress were largely achieved. One site presented figures and status on progress in bimonthly meetings of the Municipal User Council.

Only one site had representatives from the user and the carer organisations as part of the local working group. In that site, coordinators emphasised how valuable this involvement had been across strategy areas:


*“One of our successes is that we’ve achieved a unique collaboration with the user organizations at the system level. They have been very useful and contributed a lot in terms of service development…We have been fortunate in that these organizations have competent representatives willing to spend time on this… There has been mutual learning.”*
M5

The interviews with representatives from user and carer organisations reflected a less optimistic view of the success of user involvement. In their experience, there had been more information sharing than real involvement. As a result, user/carer participants experienced limited opportunities for input or influence, including involvement in evaluative work.


*“Information has not been a problem, but it hasn’t been that easy to get involved in the ‘bigger’ things such as the development of social activities or accommodation issues. As far as involvement in the evaluation of the intervention is concerned, we can’t say we’ve really participated in that. Involvement worked fine the first year, but nothing much has happened in the last six months.”*
M4

Some reported that information was often communicated through the Municipal User Council, which meant it did not always reach them. Whether our participants from user organisations had received invitations to events or training courses varied, and some said they only received these when specifically requesting it. In sites where contact between user/carer organisations and services was in place prior to the intervention, more “real” involvement was reported, for instance, through involvement in kick-off seminars and implementation meetings and participation in training courses.

Specific actions to facilitate individuals’ involvement in their own care included joint crisis plans and post-incidence reviews. These were seen to inform professionals across services so that, for instance, emergency services, took the agreed approach during a crisis. Joint crisis plans could be difficult to set up due to time constraints, difficulties in collaborating with specialist services, and sometimes because the digital systems across services did not interact. The template developed for post-incidence reviews was found too comprehensive and was therefore adapted to local needs. Where processes for setting up such reviews were established, these were seen as a valuable method for involving the person in question, and for improved service collaboration in their care.


*“Post-incidence reviews were one of the things that were new to us. It is now well established for those in our services who have been involuntarily admitted, but they can be challenging to achieve… Every case with a post-incidence review kind of becomes a quality improvement job regarding how those involved work and collaborate. It gives insight and promotes cooperation for all those involved.”*
M5

Reviews could be difficult to arrange if hospital wards failed to notify municipal services of someone being admitted or discharged. It was also noted that some municipal staff perceived people struggling with substance abuse or without the stability of good housing as “difficult to motivate” for involvement in joint planning or reviews, which could impede these actions.

Only one site had employed peer workers who were involved in “hands-on” work with those at risk of involuntary care, and this was in place before the intervention. While other sites had peer workers in different parts of service delivery, they failed to achieve this specifically targeting those at risk. Services also struggled to involve family members in post-incidence reviews, and there were no routines for involving family members if the person involuntary admitted did not want it.

#### 3.2.4. Strategy Area 3: Competence Development

This strategy area had tangible actions in the form of identified training courses that were made available. The courses were described as valuable in promoting new ways of thinking in complex organizations. Some coordinators expressed that there had not previously been a shared perspective about how the principles of recovery had implications for their work towards the target group, and that the course consolidated more of a united approach going forward. The course on trauma-informed care was frequently mentioned by staff and user/carer representatives as having been particularly valuable. It had identified ways of working that felt safe even when service users displayed fear or aggression.


*“They learned that a person might appear angry when the user actually felt scared and unsafe. Awareness of this made them focus on making the user safe instead of being afraid of them, and this meant they got a much better relationship with her.”*
M4

This course included a module on implementation aimed at managers. This was experienced as particularly relevant, and with transfer value across the intervention. There was consensus that this should have taken place earlier in the process and be open to all stakeholders. The online course on assessing decision-making capacity was appreciated by coordinators. They were of the opinion, however, that GPs and A&E doctors did not have the time neither to take the course nor conduct the thorough assessments that the course promoted. User and carer representatives expressed that this course was not sufficiently oriented towards their needs and that a more tailored approach would be preferable.

Training courses and actions for competency development were experienced as promoting engagement and motivation for the intervention. Due to pandemic restrictions, the courses were delivered online. This helped to keep the costs of participation down, which made it accessible to a wider group. It precluded, however, participation of those without necessary equipment or internet access at home. Some staff were only permitted to log on from an institutional computer at work, which could be difficult during lockdown.

#### 3.2.5. Strategy Area 4: Collaboration across Primary and Specialist Care Levels

In some cases, structures for collaborative work across service levels were in place prior to the intervention, such as regular shared management meetings. Nonetheless, this strategy area was rated lowest in terms of the degree of implemented actions. Coordinators reported an overall impression that specialist services often showed disinterest in collaboration initiated at the primary level:


*“We didn’t experience an enormous amount of enthusiasm among managers in the local specialist services when we informed them about the intervention. It was kind of “you do your work and we’ll do ours”… It’s not that easy to gain enthusiasm and commitment in specialist services to projects initiated by the municipality.”*
M3

Progress was reported, however, during the intervention period, such as the establishment of regular meeting points following the local kick-off seminar:


*“We got in place regular monthly meetings between municipal services, the FACT team, the wards, and the CMHC. Since then, it’s run smoothly. It’s been very good and has transfer-value to other parts of service delivery as well. Things get easier when we meet. I sort of notice a before-and-after effect. We’d been working for a while prior to ReCoN to get better collaboration, so when we got this joint meeting in place, it became the high point of what we’d been trying to achieve over time.”*
M1

Such meeting points enabled the sharing of success stories across service levels, which was seen as demonstrating what was available in other services, and achievable through continued collaborative work.

Despite good managerial collaboration, the action of improving collaboration and information exchange when someone was referred to, in, or being discharged from involuntary care, was difficult to achieve fully. Many at the municipal level experienced not acquiring such information, which precluded them from, for example, being present at discharge meetings. The brevity of involuntary admissions (sometimes less than 24 h) could make this action difficult for practical reasons:


*“We still have people being discharged without the municipality having been notified that they were admitted. This could be because the user doesn’t want the municipality involved, or that the admission is very brief and that there is no time to contact the municipality before discharge.”*
M3

The coordinators expressed, however, that the action of data sharing on involuntary admissions (strategy area 1) forged relationships that paved the way for collaboration in individual cases. When specialist services experienced what municipal services could offer those with frequent admissions, they seemed more prepared to involve them in pending discharges. Moreover, that municipal services now focused on, and took a degree of responsibility for, preventing the use of compulsory care meant they felt enabled to be more proactive vis-à-vis specialist services:


*“Now we in municipal services also have ownership as regards involuntary care… Now we dare ask questions and make demands. Something happened with the way we work on these cases and how we cooperate on them.”*
M5

In some regards, the Covid lockdown had made some aspects of joint work easier, because digital meetings suddenly became the norm. This was especially valuable where travel distance previously had been perceived as a barrier:


*“Covid was a driver for establishing structured, regular collaboration with specialist services. It’s much easier to set up joint digital meetings than getting everyone to attend in person. Due to Covid, all services got online and developed a culture for online meetings.”*
M1

Digitalisation could also come at a cost, however. Suboptimal IT equipment meant some struggled with communication, and sometimes digital tools, such as patient records or messaging services, were incompatible, which hampered collaboration.

#### 3.2.6. Strategy Area 5: Collaboration within the Primary Care Level

Overall, the intervention was considered to have promoted local collaboration for those at risk of involuntary admissions. The co-creating workshops and the initial kick-off seminars helped form or cement relationships and enhance collaboration between the municipal services present: mental health, the Labour and Welfare Office, the police, CMMOs, and housing services. The training courses (in Strategy Area 3) also facilitated mutual learning across services and user organisations. The action of having the prevention of involuntary admissions as part of the agenda of interagency collaboration was enabled by how the intervention was anchored at the managerial level:


*“Sharing experiences and success stories across services also provides knowledge about each other, and everyone becomes aware of other services.”*
M5

The action of compiling information to GPs and A&E medics about which municipal services they could obtain for those at risk in order to avoid referring to involuntary admissions was achieved and disseminated digitally. Coordinators were uncertain, however, to what degree the GPs made use of it. Other actions involving GPs were hampered by the pandemic. Given their organisational position, it had originally been agreed that the CMMOs would form a link to local GPs, for instance by letting municipal mental health service leads take part in their regular meetings with GPs in their area. Due to the CMMOs responsibility for COVID-19 infection control, this did not happen. GPs were involved less than anticipated, even when considering the effect of the pandemic. Some mentioned that securing GPs involvement in collaborative work was a longstanding issue. It was suggested that GPs did not consider coercion reduction as part of their remit, while others explained it with GPs’ workload, which made them prioritise hard what to be involved in:


*“GPs have too much—they are ‘vaccinated’ against things coming from the outside. They have an information overload aimed at them.”*
M2

To varying extent, GPs took part in regular ‘collaboration meetings’ (‘ansvarsgruppe’ in Norwegian), which are designed to bring the appropriate professionals together with the person who is unwell to plan and coordinate services that meet their needs. This might include people from specialist and primary services, the GP, the Labour and Welfare Officer, and sometimes family members. All municipalities had arrangements for such meetings in place prior to the intervention and maintained them throughout. This was perceived to provide opportunities to specifically focus on how to avoid or address individual psychiatric crises.

#### 3.2.7. Strategy Area 6: Tailoring Individual Services

In the early stage of the intervention period, coordinators expressed that many of the actions in this strategy area were already in place (including the case collaboration meetings just mentioned) and considered existing routines for follow-up service based on individual needs as generally good. For instance, there were existing collaborations with the local Labour and Welfare Office for joint work in individual cases. Some of this work came to a halt during the pandemic, however. This was also the case regarding supporting people to take part in leisure activities: the pandemic restricted everyone’s movement and forced many activity centres to close. In the periods this support was possible, it had been difficult to recruit sufficient sessional support workers. Turnover, sick leave, and use of temporary staff among those in direct contact with those who might benefit from these services, also affected these actions because a lack of continuity in contact with an individual made it harder to detect their needs and tailor services accordingly.

As the intervention progressed, coordinators expressed how they gradually implemented the action of reconsidering individual accommodation needs following admissions. While they had some scope for matching accommodation to needs, the limitations in social housing meant this could be difficult to achieve, and supplementing the municipal housing stock was not within the service’s remit or budget.

In some sites, the action of having local crisis beds was in place before the intervention, and one municipality created a crisis bed in their residential mental health facility. It is a statutory requirement for municipalities to have beds in general Municipal Acute Units, but these are rarely used for mental health crises. At least one municipality started to use such beds for this purpose during the intervention period, and in one site one such bed was dedicated to mental health. The impression was that using local crisis beds prevented involuntary admissions:


*“We’ve had about 100 bed days at the Municipal Acute Unit, for 21 users. We see that a few of them are referred on, which is good. This has slowed the deterioration of symptoms. In some cases, we’ve also used Municipal Acute Units before a rehab placement to ensure that the person makes it there.”*
M3

One municipality had not used its newly established crisis beds during the intervention. They speculated whether this was because they now prevented crises from happening, or whether GPs and A&E services were unaware of this option.

It became clear throughout the period that actions in other strategy areas facilitated services’ ability to meet individual needs better. As already mentioned, the use of service data to monitor involuntary admissions alerted municipal services to those at risk but not previously known to them. It also helped identify and target support for those with repeated admissions.


*“Getting figures from the hospital has helped us identify about five people with frequent involuntary admissions. We have worked extensively with these people and we have found individualised, good solutions for them so that they have managed to break the patterns of admissions. What has helped have primarily been (our) increased awareness; collaboration with the hospital; more tailored living situation, and; better cooperation with the FACT team.”*
M1

Coordinators also describe that actions across strategy areas had resulted in a change of focus and a more holistic approach to individual needs:


*“This young person, with serious mental illness and also somatic disease, has shifted from regular involuntary admissions to voluntary ones. The structured work with joint crisis plans and post-incident reviews has been among the success factors here. … We have established a stable team of good professionals around this person, built relationships, and things have improved. This also applies to the collaboration with the hospital and CHMC, although this varies a bit…We are now looking at this person’s housing situation and whether it contributes to making them worse rather than promoting recovery … We need to upgrade both this person’s housing and the competence of the professionals involved.”*
M5

## 4. Discussion

The ReCoN intervention consists of six strategy areas with a total of 53 actions (Table 1). Across the sites, some actions needed to be initiated, and others to be maintained. This was ambitious in scope, especially as no extra resources were available. Nonetheless, all sites implemented, either fully or partially, the majority of the actions. As shown, study participants perceived overall the outcomes of the intervention as positive, and that it had impacted pathways of those at risk of involuntary admission. Whether the changed ways of working will reduce the level of involuntary admissions, as measured by the cluster-RCT, remains to be seen.

To push implementation processes in the desired direction, drawing attention to facilitators and barriers is recommended [29]. The regular implementation meetings and progress interviews might therefore have been facilitating progress. Our thematic analysis identified a range of facilitators and barriers. External factors with impact across the intervention include most notably the COVID-19 pandemic restrictions. Incompatibility of patient records and IT-systems were also a challenge, which is not uncommon in interventions spanning organisational boundaries [15].

The different strategy areas depended to some extent on each other for successful implementation: progress in one area helped progress in others, while a lack of success could impede actions elsewhere. For instance, good leadership and appropriate organisational anchoring facilitated actions across strategy areas, and its absence was identified as a barrier. Similarly, the active use of data, post-incident reviews, joint crisis plans, and improved collaboration, which formed part of different strategy areas, were all seen to improve the ability to tailor services to individual needs (area 6).

We believe the intervention’s interconnectedness stems from the co-production process [13], which meant it was founded on extensive experiences of local collaborative work, even if the topic of coercion reduction was new to municipal services. The six strategy areas stakeholders chose to focus on are known in the literature on collaborative care for being tricky [15,30,31]. As such, stakeholders perceived that headway needed to be made across these areas if the pathways of those at risk should be steered away from involuntary admissions. The co-creation process meant that the municipal mental health services that were to implement the intervention, determined its content together with local collaborators. As also reported for other healthcare interventions [32], this can ensure face validity, contextual fit, and acceptability, and enable better implementation [13,33].

A recent systematic review of interventions to reduce involuntary care (all from inpatient settings) found eight outcomes on which successful implementation is assessed. These are that the intervention is experienced as *acceptable* and *appropriate*, that it is *feasible* in practice, that it is *adopted*, and with *fidelity* to plans. *Penetration* in and across services is also important for its *sustainability* [14]. Our results indicate that substantial progress was made on all these aspects in the implementation of the ReCoN intervention, even if there were concerns about the extent of its penetration in collaborating services. Also, longer-term sustainability needs to be assessed at a later stage.

The interconnected nature of strategy areas created a coherence in the intervention that we believe has contributed to its implementation, and that can also inform other complex health systems interventions [34]. We next discuss four key areas where our findings might have particular relevance, before making recommendations for future implementation of the ReCoN-intervention.

### 4.1. Establishing and Maintaining a Shared Agenda: Leadership and Anchoring

Joint working by organisations with different remits, targets, and funding arrangements are known to be difficult [15] and individual organisations tend to shift focus when new policies or targets are imposed on them. Local collaboration has its challenges, and getting GPs on board is commonly reported as problematic [35]. Collaborative work between primary and specialist services is often marked by traditions of hierarchy and power differentials [36]. Our participants reported that establishing structures and arenas for collaboration, and simply getting to know one another could enable improved joint working. This was facilitated by structured implementation events to learn across sites and training courses in which people from different stakeholder groups took part.

Leadership and collaboration among service managers were highlighted as key to fostering a shared agenda and keeping everyone focused over time. The importance of leadership at the appropriate organisational level is a common finding in evaluations of healthcare interventions [37,38,39]. To achieve this, our participants suggested the need for anchoring at a high enough level to exert influence on service organisation internally and have clout in collaborating services while also ensuring ‘hands-on’ experience with the target population needed for implementation ‘on the ground’. Coordinated anchoring at different organisational levels might therefore be needed.

### 4.2. Stimulating a Culture for Change

Implementing change to intervene in clinical pathways, in this case, to prevent involuntary admissions, requires a ‘readiness for change’ [40] to organisational ethos or cultures [41]. Our participants expressed that a prolonged planning stage to prepare for such change might be needed, and training all stakeholder groups in implementation theory during the planning stage was a specific suggestion. Cultural change might also develop from changed ways of working. For instance, introducing the use of routine hospital data to keep track of those involuntarily admitted or ready for discharge had made municipal services more oriented towards those with experience of compulsory care and their needs during transitions. Also, the practice of obtaining data fostered relationships that enabled shared agendas for meeting those needs. Participants indicated that the intervention had instigated attitudinal change in that professionals at the municipal level now displayed awareness of their opportunities for preventing involuntary admissions and a degree of ownership of this agenda.

### 4.3. Clarifying the Role of User Involvement

While user and carer representatives took part in co-production and monitoring work, there were discrepant reports on their involvement in the day-to-day running of the intervention: while the coordinators from the mental health services assessed this involvement to be relatively good, user representatives themselves described it as marginal, and that they were more often informed than involved. Turnover in the local user/carer organisations could have impacted involvement. As few of their representatives had personal experience of involuntary care, this could have affected how the local group prioritized their work. As we explain elsewhere, user involvement related to involuntary care is demanding, and it is difficult to recruit persons with lived experience of compulsion to take part over time [42]. Expecting such involvement in an unpaid capacity might be unrealistic.

Appraisals of user involvement will vary with the understanding of the nature and purpose of how such involvement. Studies have found that professionals might consider the sharing of information as involvement, while from other perspectives, proper involvement might require shared agenda-setting and decision-making [43]. As such, the different assessments of user involvement in our study could reflect immature practices and a lack of agreement on its potential contribution. It might also reflect differing views of the value of experiential knowledge vis-a-vis professional or clinical knowledge [31]. A lack of such recognition might also in part explain why only one site achieved having peer workers with personal experience of involuntary care in the day-to-day operational work with people at risk. Securing solid mechanisms for user involvement might require both leadership and a culture for change [44].

### 4.4. Supporting Implementation and Intervention Sustainability

The structured implementation plan, which included continuous input and the presence of the research team, was highlighted by participants as key to implementation. These activities revolved around problem-solving across sites. In addition, they served as regular reminders, in the context of hectic work schedules and competing demands, of the intervention’s ambitions and activities. External support has been found to improve the implementation of a variety of complex interventions, and through that contribute to overcoming organisational barriers, resistance to change, and increasing intervention effect [45,46]. This also applies to mental health service interventions [47], including those that directly support policies or other steering mechanisms [48,49].

Dependence on external support affects the sustainability of interventions. The extent and nature of support to ensure new ways of working are mainstreamed might not be apparent until intervention periods are finished. While our study was designed so that services participated within existing resources, the research team’s contribution was covered by time-limited grant money. We believe the implementation support provided in our study, as detailed in Figure 1, could be taken on by the services themselves. This would require that this function be assigned to members of staff with dedicated time and managerial support. How to achieve this in practice should be included in future studies, including how it should be budgeted for [50].

### 4.5. How the Intervention Should Be Modified or Adapted in Future Implementation

Our study demonstrates that it is feasible to implement a complex intervention to reduce involuntary care in a primary mental health care system offering ‘general support’ [10] such as those providing long-term care for people with severe mental health problems in Norwegian municipalities. We believe the co-creation process that ensured local ownership and a good fit to local practice, was essential to achieve this. We recommend that future implementation at the outset ensure there is ownership in the local setting for the agenda of preventing involuntary admissions, and that adaptations within the ReCoN framework reflect local service configurations. In addition, we believe the following points should be taken into consideration.

First, an enhanced planning stage should take place prior to implementation. The allocation of leadership and coordination roles needs careful planning in the context of existing partnerships and local services. Necessary training courses, tailored to different stakeholder groups, should be scheduled for this stage. This might be particularly important regarding training in implementation, but also to maximise the impact of mechanisms such as joint crisis plans and post-incidence reviews.

Second, a shared understanding of what ‘user involvement’ entails should be established. The potential for how such involvement might maximise the intervention should be clarified, and the realisation of that potential adequately planned for. To secure involvement over time, appropriate payment and reimbursement of expenses should be budgeted for.

Third, a realistic level of ambition should be set for what is achievable within timeframes, organisational remits, and financial resources. While most of the actions were implemented in this study, some, such as employing additional peer workers or adding to housing stocks, have budgetary implications that must be addressed at a high level and in competition with the needs of other local services.

Fourth, what is needed in terms of implementation support should be clarified. Whether this function should be provided externally or integrated into the participating services, it should be clearly defined and allocated sufficient resources. To achieve this, there might be a case for including established tools to plan for, monitor, and evaluate implementation [40].

### 4.6. Strengths and Limitations

We applied a qualitative methodology, which is suitable for investigating processes, complexity, and nuances and is recommended in the field of implementation science [51,52]. Our methodology precludes, however, the numeric generalisation of results. IW, TLH, TH, and SHK facilitated the co-production and implementation strategy and conducted progress interviews. Such closeness gives unique insight but can also introduce bias. We attempted to remedy this by involving an additional analyst (JR), who had not taken part in these activities. The scoring of the degree of implementation used qualitative data, so can only be taken to represent impressions of trends and should be interpreted with caution. As we did not continue to assess an action once it had been considered fully implemented, we cannot rule out that some actions did not proceed over time. The use of a standardised fidelity scale or implementation tools could have provided a more robust measure of implementation degree. The structured and regular monitoring interviews with coordinators ensured that we obtained information from those closest involved in implementation. Their views may differ, however, from those of other stakeholders. The experience of user and carer representatives of having been insufficiently involved to properly assess the implementation of many actions means that their perspectives are lacking in parts of the analysis, including their impression of the degree of implementation of a number of actions. User involvement through two national organisations formed part of the design: a different approach might have enhanced this part of the work. Also, better representation from local collaborating services in the evaluation of the implementation could have offered further perspectives. Our data might be open to additional interpretations. The analysis was performed blind to the pending results from the RCT.

## 5. Conclusions

Our investigation shows that it is feasible to implement a complex intervention designed to reduce the use of involuntary admissions in general support services, such as Norwegian primary mental health services. The majority of the 53 actions were implemented in all five sites, which we believe was due to the co-creating process and the willingness of stakeholders to address tricky areas of collaborative work. The analysis of facilitators and barriers shows a high degree of interconnectedness in that progress (or lack thereof) in one strategy area affected other areas. Future implementation should ensure the intervention is adapted to, and embedded in the local service context; pay more attention to planning and training; clarify the role and contribution of service user and carer involvement, and pay close attention to the need for implementation support.

The feasibility and penetration of the ReCoN intervention are highly relevant to national and international policy for reduced use of involuntary care [1,4,14] because it shows there may be potential to work towards this aim through collaborative, ongoing general support services [10] when people live at home. Should the pending RCT result of the intervention show positive outcomes, there will be a case for rethinking policy in his area, including the way in which responsibilities and resources are distributed. This would require a culture change not only in services on the ground but also at the level of health authorities and policymakers.

## Figures and Tables

**Figure 2 healthcare-12-00786-f002:**
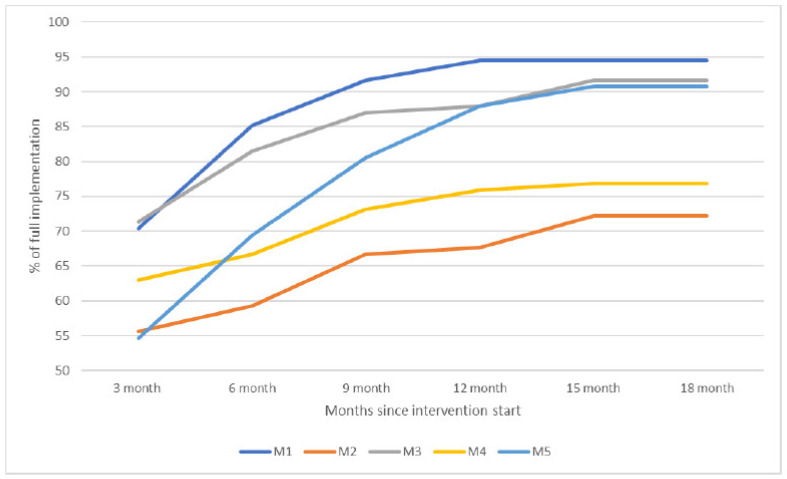
Degree of implementation in the ReCoN intervention.

**Table 1 healthcare-12-00786-t001:** The ReCoN intervention.

Strategy Area 1: Management
1.1. Management anchoring
Management anchoring in relevant organisations/services in the municipality
Appoint a minimum of two project coordinators
Anchoring support for the primary mental health services’ implementation of the intervention in relevant collaborating services at both primary and specialist care levels
Management support, facilitation, and prioritisation to enable implementation of actions
1.2. Data monitoring—use of data in service development
Develop a plan for how to collect and use data: what, how, who should collect it, how often, and how it should be used
Establish routines for registering data based on the above data monitoring
Establish routines for utilising the data based on the data monitoring plan, e.g., every unit/service examines and evaluates the data for their unit/service in their regular staff meeting
Evaluating progress over time
1.3. Continuous service improvement
Establish routines for documenting situations that led to referrals for involuntary admissions and for “success stories” where this was prevented
Examine and evaluate all situations that lead to referrals to involuntary admissions and “success stories” where this was prevented every third month
Communicate the results from such evaluative work and apply it in service improvement work
**Strategy Area 2: Involving Persons with Lived Experience and Family Carers**
2.1. Involving persons with lived experience and family carers at the organisational level
Invite local user and carer organisations, Municipal User Boards, or other user representatives to participate in the intervention
User representatives participate in evaluative work
User representatives participate in the evaluation and reflection on referrals to involuntary admissions every third month
2.2. Post-incident review
Establish routines for offering those referred to or discharged from an involuntary admission a post-incidence review
Conduct post-incident reviews after each involuntary admission
Conduct post-incident reviews after each referral for involuntary admission
Establish routines for contacting family carers of individuals who have been involuntarily admitted or referred to such admission to check if they want a post-incident review.
Contact and conduct a post-incident review with family carers who were not part of such reviews of those referred to or involuntarily admitted
2.3. Joint crisis plan
Examine whether all those involuntarily admitted during the last 12 months, or who are deemed to be at risk for such admission, have a joint crisis plan, and produce a joint crisis plan for those who do not have one
Establish routines to secure that up-to-date joint crisis plans are anchored in and available to involved services (consent needed)
Prepare a joint crisis plan for all those discharged from an involuntary admission.
Update the joint crisis plan, together with the individual, at least every sixth month, and always after a crisis or an involuntary admission
2.4. Peer worker
The municipal mental health services have peer worker(s) with relevant lived experience who work closely with individuals at risk of involuntary admission.
**Strategy Area 3: Competence Development**
3.1. Recovery-oriented framework
Have an explicit principle of a recovery-oriented foundation for service provision
3.2. Competence-building measures
All relevant services/staff participate in a three-hour digital course in assessing decision-making capacity
People with a mental illness and their family carers and network could be encouraged to undertake the digital course in assessing decision-making capacity
A one-day training course on recovery-orientated services
Training course in trauma-informed care
**Strategy Area 4: Collaboration across Primary and Specialist Care Levels**
4.1. Collaboration when assessing someone for involuntary admission
Getting support from the specialist mental health service provided when assessing individuals’ capacity to consent to treatment, especially when there is uncertainty
Collaborating with specialist services on finding alternatives to involuntary admissions
Establish routines for primary mental health services to be notified when an individual is referred to involuntary admission, but such an admission is not established
4.2. Collaboration during and following involuntary admission
Collaborate with specialist mental health care on conducting post-incident reviews after each involuntary admission
Collaborate on preparing joint crisis plans during admissions (Can form part of the “discharge meeting” below)
The relevant primary care services participate in discharge meetings from involuntary admissions
4.3. Joint meeting points
Joint evaluation meetings at the management level. Can take place on existing meeting areas at the managerial level
Primary and specialist services participate in ‘case collaboration meetings’ for individuals at risk of involuntary admissions
**Strategy Area 5: Collaboration within the Primary Care Level**
5.1. Collaboration between GPs/A&E services and the primary mental health services
Familiarise local GPs with the intervention in collaboration with the chief municipal medical officer
Provide an overview of available primary care level services that might provide alternatives to involuntary admission
Communicate/disseminate this overview to all collaborating GPs and emergency medical services
5.2. Joint meeting points
Establish ‘case management meetings’ for all individuals at risk of involuntary admissions if these are not already in place
Have reduction of involuntary admissions on the agenda of meetings at the managerial level
Arrange a kick-off seminar of the intervention period with relevant services from primary services and other stakeholders such as user and carer organisations and specialist services.
Arrange a day for joint professional development on recovery-orientation
**Strategy Area 6: Tailoring Individual Services**
6.1. Individually tailored accommodation
Assess, during discharge meetings/post-incidence reviews, whether someone’s current accommodation is appropriate and stable
6.2. Primary level crisis or short-term placement
Establish primary crisis retreats or short-term institutional places, or, if already present, evaluate how such beds are utilised
Use such services for those in the target group when the individual and services assess it to be appropriate
6.3. Support for meaningful everyday lives
Establish collaboration with the Labour and Welfare Organisation for those in the target group who are eligible for and wish to receive these services
Assess the need for support, offered by the municipality, to engage in social interaction and leisure activities
Assist individuals in the target group to seek economic support from the municipality to participate in activities they want and need
Assist individuals in the target group with transport to and from activities if they need and wish it
Assess needs and consider appropriate actions to help individuals manage their sleep, diet, exercise, and money
The case manager is involved in an individual’s interaction with his/her GP follow-up regarding medication

Translated from the Norwegian implementation manual (Hatling, T.; Husum, T.L.; Kjus, S.H.H.; Wormdahl, I., 2020 [25]) by the authors and reproduced with permission.

## Data Availability

The qualitative data on which this manuscript is based will not be publicly available as it is impossible to remove all identifiable features in the interview transcripts to ensure interviewees, and the clinical encounters they discuss, cannot be identified.

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
