# Peer review of "Something Happened with the Way We Work: Evaluating the Implementation of the Reducing Coercion in Norway (ReCoN) Intervention in Primary Mental Health Care"

_healthcare, 2024, doi:10.3390/healthcare12070786_

Round 1

Reviewer 1 Report

Comments and Suggestions for Authors

The manuscript was very interesting and part of bigger project. This manuscript consentrated to implementation and waiting for future the results and numbers of effectedness of th interventions.

Page 2 you tell that you made cluster-randomised controlled trial, but I didn't find the results of that? Did you only presented the results of implementation to five randomised municipalities?

I hope you could move The service and legal context before The intervention and implementation plan.

Methodology: What are the research questions? In results you mentioned that there were problems between the primary and special level, but after joined crisis plans and post-incidence reviews was increased the understanding. How does the special level participated to the research?

Reviewer 2 Report

Comments and Suggestions for Authors

The study discusses the implementation of the ReCoN intervention, aimed at reducing involuntary admissions in Norwegian primary mental health services. Despite facing challenges, the intervention, comprising six strategy areas with 53 actions, was largely implemented across various sites. The study identifies facilitators such as leadership and collaborative efforts, but also notes barriers like external factors and IT system challenges. The role of user involvement is highlighted, and recommendations include a shared agenda, enhanced planning, and ongoing support for sustainability. The feasibility of the intervention is acknowledged, pending the results of a randomized controlled trial. It is an orginal and valuable study that would contribute relevant literature.

  1. Including the details of the study and the case recruitment process in the introduction section in the method section will provide a better perception of the study. This part should be integrated into method.

  2. The description of the ReCoN intervention outlines a complex and ambitious plan with multiple components. The complexity could hinder effective implementation, especially in real-world, resource-constrained settings. This should be mentioned as a limitation and discussed in relevant sections.

  3. The extension of the intervention period due to Covid-19 may raise concerns about the study's ability to adhere to the original timeline and may introduce external factors that could potentially influence the outcomes. This problem should be mentioned and add as a limtation.

  4. The reliance on extensive notes from progress interviews and a qualitative evaluation seminar might be seen as a limitation, as the absence of quantitative data may compromise the robustness of the analysis. This could raise concerns about the objectivity and generalizability of the findings. Please add this as a limitation

  5. The categorization of the degree of implementation into fully implemented, partly implemented, or not implemented relies on subjective judgment. This introduces a potential bias and lack of objectivity. The scoring system should be detailed and procedures in case of inconsistency should be described.

  6. The reliance on a single end-of-intervention evaluation seminar with coordinators and representatives may not provide a comprehensive understanding of the intervention's impact. A more frequent and diverse set of evaluation sessions could have offered more detailed results on the ongoing implementation process. This could be offered as a suggestion for future studies.

  7. The exclusion of user representatives' perspectives in scoring actions, except those related to user involvement, may limit the validity of the evaluation. Please discuss it.

  8. None of the municipalities completed all actions, indicating a lack of comprehensive success. This might cause concerns about the effectiveness of the intervention and its ability to address all relevant aspects of the problem. Please elaborate on it.

  9. The explanation for slower implementation, particularly attributing it to Covid-19 restrictions, suggests that external factors significantly influenced the study. Relying on external circumstances as a primary explanation brings questions about the intervention’s efficacy under real-world conditions. What is your future suggestions to eliminate it in the future?

  10. While the study highlights the feasibility of the ReCoN intervention, it emphasizes the pending results of a randomized controlled trial (RCT). Authors should mention about their expectations of the outcome of mentioned study.

Round 2

Reviewer 1 Report

Comments and Suggestions for Authors

Thank you for improving the manuscript. 

Author Response

Reply to Academic Editors notes:

  1. I think that you have responded to the critiques raised by reviewers.
    I think the underlying demonstration and quantitative evaluation will be extremely important and the fact that it has not been completed should also be noted in the abstract.

Authors reply: Thank you for the comments! We have now made it clearer in the abstract that results from the RCT is not published yet as follows:

«The intervention was implemented in five municipalities and is being tested in a cluster randomized control trial, which is yet to be published. The present study evaluates the implementation process in the five intervention municipalities To assess how the intervention was executed, we report on how its different elements were implemented, and what helped or hindered implementation. Methods:»

  1. I think there is room for your team to think a bit more and perhaps address in the discussion section, that without data on impacts/effectiveness, it is hard to know whether the positive implementation experience was or was not an element in a successful program..

Authors reply: Of course, the participants positive experiences of the implementation may have been a result of taken part of the study or by other factors or reasons than the success of the intervention., The question of whether the intervention also had a positive effect on the primary outcome measures of the study had, remains to be seen when these have been analysed and published. To address you concern we have added the following at the beginning of the Discussion, in addition to the comment on the pending RCT that already is in the Conclusion:

The ReCoN intervention consists of six strategy areas with a total of 53 actions (Table 1). Across the sites, some actions needed to be initiated, others to be maintained. This was ambitious in scope, especially as no extra resources were available. Nonetheless, all sites implemented, either fully or partially, the majority of the actions. As shown, study participants perceived overall the outcomes of the intervention as positive, and that it had  impacted pathways of those at risk of involuntary admission. Whether the changed ways of working will reduce the level of involuntary admissions, as measured by the cluster-RCT, remains to be seen.

  1. I think the paper really has 2 research questions...the 3rd question is about how the program should be modified in future applications. As far as I can tell, you did not request guidance or input from the 53 respondents in the final seminar or in the field notes from interviews on this question----rather you draw these implications from what they reported experiencing with implementation. You studied questions 1 and 2 to see if you could identify lessons for future application of the model or to find ways to potentially explain significant findings from the outcomes evaluation. The discussion of lessons for future implementation is interesting but it seems to represent what the authors learned from this experience about future adaptations of the model not from analysis of qualitative data. I would eliminate the 3rd question but maintain the discussion of implementation lessons.

Authors reply: We did address this question specifically with the participants at several occasions, during the interviews with the coordinators and at the last meeting which marked the end of the intervention period. Our discussion of possible modification is, however, also shaped by the general literature (which is why we placed in in Discussion), so we have now made a small change to the aims of the study to make it more precise. We have made a similar change to the Abstract –see changes on the first comment.. We hope this will answer your concern.

Specifically, our objectives were to investigate (i) to what extent the different elements of the intervention were implemented, (ii) what helped or hindered implementation. Based on these results and the wider literature we then consider, in the Discussion section, how the intervention should be modified or adapted in future implementation

You were asked by reviewers to speak about the limits of your data and you additions to this section suggest that you did not fully respond to this concern.....I do not think that this was a request to justify qualitative research but rather a request to express that the seminar post-study and field notes represent public i participants in implementation and not systematic interview and directly observable data on degree of implementation or systematic data collection around the challenges in implementing each component.

Authors reply: Our monitoring interviews were in fact regular and fairly systematic. We agree, however, as already stated in the ‘limitation’ section there are several levels in this study where subjective interpretations may have had influence We tried to minimize this by doing the analysis and the interpretation together with several researchers, and adding a senior researcher in the last phase of the interpretation who was not involved in the evaluation herself.  We have alsl added a note that more structured approach to measurement might have been an advantage. We hope this answers your concern: 

The scoring of the degree of implementation used qualitative data, so can only be taken to represent impressions of trends and should be interpreted with caution. As we did not continue to assess an action once it had been considered fully implemented, we cannot rule out that some actions did not proceed over time. The use of standardised fidelity scale or implementation tools could have provided a more robust measure of implementation degree. The structured and regular monitoring interviews with coordinators ensured that we obtained information from those closest involved in implementation
